# Effects of Chemical Priming on the Germination of the Ornamental Halophyte *Lobularia maritima* under NaCl Salinity

**Iman Zammali, Amira Dabbous, Seria Youssef and Karim Ben Hamed \***

Laboratory of Extremophile Plants, Centre of Biotechnology of Borj Cedria, P.O. Box 901, Hammam Lif 2050, Tunisia; imanzammeli9@gmail.com (I.Z.); amiradabousbenali@gmail.com (A.D.); seriayoussef93@gmaill.com (S.Y.)

**\*** Correspondence: karim.benhamed@cbbc.rnrt.tn

**Abstract:** *Lobularia maritima* or sweet alyssum (Brassicaceae) is an annual littoral halophyte, naturally thriving on sandy beaches. In addition to its obvious interest as a naturally salt-tolerant plant, this species is mainly cultivated as an ornamental plant in many countries. Laboratory experiments were carried out to assess the effects of salinity on seed germination and on germination recovery from the effects of saline conditions after transfer to distilled water. Seed germination responses were determined at 0, 50, 100, 200 and 300 mM NaCl. Salt (NaCl) does not affect the germination of *L. maritima* if applied at a moderate dose of 50 mM. For higher concentrations of NaCl, there is a decrease in the germination rate at 100 and 200 mM NaCl or even a total inhibition of germination at 300 mM. Salt lowers or inhibits germination only through osmotic effects. To improve the germination of *L. maritima* under high salinity, seeds were pretreated with $KNO_3$, thiourea, proline and salicylic acid. The germination of seeds is improved by $KNO_3$ in the presence or absence of salt, while thiourea increases the final germination without affecting the germination rate. Salicylic acid amplifies the effect of salt, while proline delays germination without stopping it completely. These findings indicate that the application of $KNO_3$ and thiourea may be used to improve seed germination of *L. maritima*, which is of great interest for cultivating this plant for landscaping purposes in saline soils.

**Keywords:** germination; sweet alyssum; landscaping; ornamental; seed priming; salt stress

## 1. Introduction

In recent years, rapid urbanization has increased the demand for landscaping in many countries. At the same time, the availability of good-quality water for gardens and landscapes is becoming increasingly restricted due to the rising demand for domestic use of the scarce fresh water resources. Consequently, landscape architects are in search of ornamental plants that perform well with saline groundwater and treated wastewater. To date, only limited systematic work has been carried out to study the tolerance of landscaping plants to higher levels of salinity. Ornamental flowering plants, such as carnations, roses, chrysanthemums, and gerberas, found around the world, albeit known for their high economic, ornamental, ecological, and edible value, are sensitive to salt stress. Recent studies showed that the salt tolerance of ornamental crops such as chrysanthemum can be improved through genetic manipulation [1,2].

Alternatively, some naturally salt-tolerant plants (halophytes), such as *Aster tripolium*, *Sesuvium portulacastrum*, *Limonium* sp., *Crithmum maritimum* and *Lobularia maritima*, can be used for landscaping and ornamental purposes, and cultivated in salt-affected areas under salinity irrigation [3]. According to the e-HALOPH database [4], many halophyte species produce attractive flowers. Cassaniti and Romano [5] investigated halophytes native to the Mediterranean region and listed 13 families with about 42 species of ornamental potential. Although many halophytes can be considered as promising ornamental crops, only little information is available in the literature regarding their cultivation techniques.

Salinity is known to impair seed germination of halophyte species and they may be as salt-sensitive as glycophytes in the early stages of their life cycle [6] Under natural conditions, seeds of halophytes are subjected to salt stress dominated by NaCl, yet, other salts can also affect seeds germination significantly due to the low osmotic potentials occurring under saline conditions or to the toxic effects of the ions [6]. Salinity induces both a reduction in the percentage of seeds germinating and a delay in the initiation of the germination process. At salinities beyond the tolerance limits of the species, complete inhibition of the germination process can occur because of: (i) inadequate imbibition; (ii) ionic toxicity; (iii) interference with metabolism; (iv) destruction of enzymes; and (v) imbalance of growth regulators [7]. Once salinity is reduced following rain, seeds of halophytes have the capacity to recover from the salinity shock and start germination for non-dormant seeds [8]. Impaired germination is partly resulting from the salt-induced reduction of the growth regulator contents in seeds, which control germination activity in most halophyte seeds [8,9]. Interestingly, seed dormancy can be broken either by physical treatments, including scarification, or by chemical treatments by the exogenous application of gibberellins, nitrogen or ethanol, which can efficiently alleviate the harmful effect of salinity on halophyte germination [10–13].

*Lobularia maritima* L. Desv., commonly known as sweet alyssum—is a species of the Brassicaceae that is distributed throughout the Mediterranean basin, where it grows in coastal zones, dunes and scrublands. This plant forms a basal rosette with prostrated-ascending flowering stems. The plant is a short-lived perennial with a lifespan of approximately three years but with a long flowering period because *L. maritima* individuals produce flowering stems more or less regularly over the completely flowering season [14]. It has been cultivated both as an ornamental plant in many countries in the world and as an insectary plant to intercrop with lettuce because alyssum flowered quickly after planting was not overly aggressive or likely to become a weed, and attracted several beneficial species, but few pest species [15,16].

In the current study, our aim was to address the following three questions: (1) is germination of *L. maritima* inhibited by an osmotic effect and/or a specific ion effect, (2) to what extent can seeds recover from exposure to high concentrations of sodium chloride, and (3) which suitable (pre)treatment that improves germination and overcomes the effects of NaCl by investigating the role of $KNO_3$, thiourea, proline and salicylic acid during the germination of *L. maritima* under increasing NaCl salinity in order to get more information on the mechanisms by which salinity may inhibit germination of this species in the natural conditions.

## 2. Materials and Methods

### 2.1. Plant Materials

Seeds of *L. maritima* were harvested on May 2019 in Tabarka, a locality close to the Mediterranean seashore, NE Tunisia (36°57′23″ N 8°45′28.5″ E). They were separated from the inflorescences and stored dry at 4 °C. They were sterilized using 70% ethanol for 5 min, 40% bleach for 10 min and washed five times with sterile water.

### 2.2. Germination Tests

Seeds were placed in tightly sealed 9 cm Petri dishes with Whatman paper # 1 moistened in 2 mL of distilled water or a test saline solution. Three replicates of 20 seeds were used for each treatment. Germination was carried out in the dark at 20 °C (incubator KRG-250 model). The dishes were checked daily (for 45 days) to count the germinated seeds. Seeds developing radicals of >1 mm length were considered germinated.

For evaluation of salinity tolerance at seedling stage, seeds were sown in Petri dishes on filter paper soaked with NaCl solution (50, 100, 200 and 300 mM). Control seeds were treated continuously with distilled water. Ungerminated salt-treated seeds were transferred to filter paper soaked with distilled water for another 7 days to determine their ability to recover from salt pretreatments. To improve *L. maritima* seed germination under

increasing NaCl concentration, seeds were soaked for 24 h in four chemical treatments, afterwards, they were washed thoroughly with distilled water, dried (to obtain the initial seed weight), and kept for germination at 25 °C in growth chamber. The impact of the external application of $KNO_3$ (10 mM), thiourea (50 mM), proline (1 mM) or salicylic acid (10 μM) on germination was tested over a range of NaCl concentrations (0, 50, 100, 200 and 300 mM). These concentrations of priming substances were selected on the basis of a preliminary trial (data not given).

### 2.3. Data Analyses

Three parameters of germination were determined: final germination percentage, germination rate and germination recovery percentage. For each Petri dish, final germination percentage was calculated as:

Final germination = (number of germinated seeds/number of sampled seeds) × 100.

The rate of germination was estimated using a modified Timson's index of germination velocity:

$$\text{Rate of germination} = \Sigma \, G/t$$

where G is the percentage of seed germination at each measurement and t is the number of measurements.

The germination recovery percentage was determined by the following formula:

$$\text{Germination recovery} = [(a - b)/(c - b)] \times 100$$

where a is the number of seeds germinated after being transferred to distilled water, b is the number of seeds germinated in saline solution, and c is the total number of seeds.

The data were subjected to one-way analysis of variance (ANOVA) to evaluate the effect of the salinity and their interactions on the final germination percentage, germination rate and recovery percentage. Tukey's test (honestly significant differences, HSD) was carried out to perform all-pairwise comparisons between individual treatments ($p < 0.05$). All statistical analyses were performed using SPSS Statistics 20 (IBM Corp., Armonk, NY, USA) software.

## 3. Results

### 3.1. Effects of Salt Treatment on Final Germination

The kinetics of germination at 50 mM NaCl are generally similar to those observed in the control medium (distilled water), except that the germination rate is higher in the presence than in the absence of salt (Figure 1). Germination does not exceed 20% at 100 mM and 200 mM NaCl. It is completely inhibited at 300 mM NaCl (Figure 1).

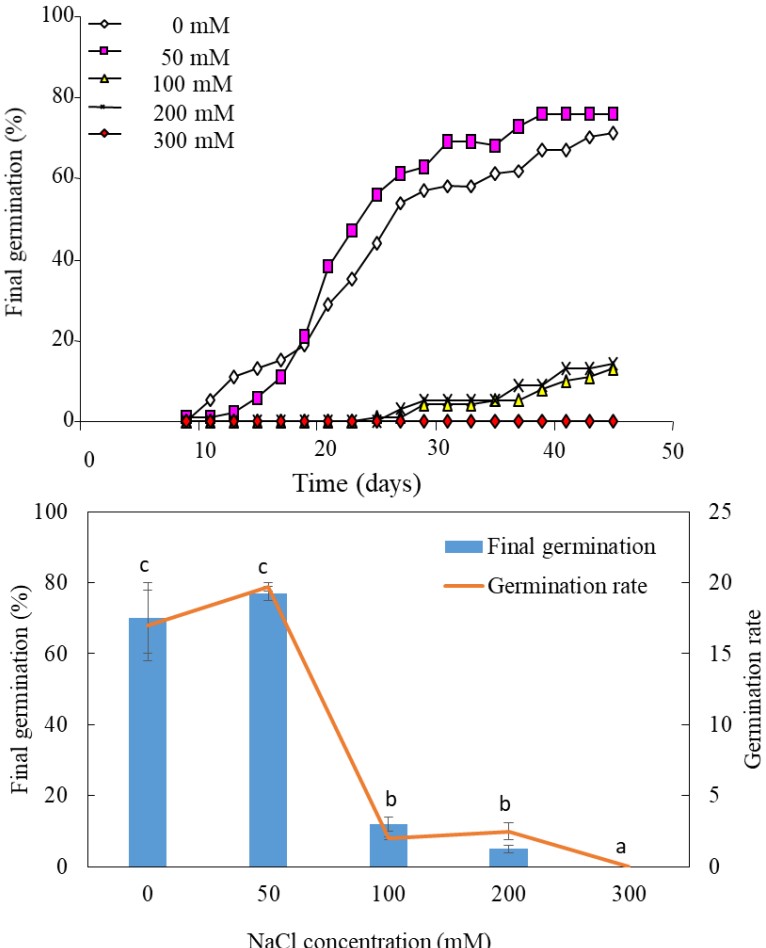

**Figure 1.** Effect of salinity on the germination of *Lobularia maritima*. Values at each level of salinity with the same letter are not significantly different (*p* < 0.05); Tukey test. Data are presented as means and standard errors of three repetitions.

### 3.2. Effects of Salt Treatment on Germination Recovery

The test of the reversibility of germination consists in transferring the seeds that have not germinated on salt medium to a control medium devoid of NaCl in order to determine the nature of the effect of salt: toxic or osmotic. Seeds that have not germinated at 100, 200 or 300 mM, once transferred to distilled water, regain their ability to germinate at rates of 80%, 65% and 50%, respectively (Figure 2). As the germination capacity of the seeds that have not undergone any treatment is of the order of 80%, it is deduced that the stay of the seeds for 45 days in the presence of salt did not affect their viability. The inhibition of germination previously observed in the presence of salt is therefore mainly of an osmotic nature. However, a toxic effect is also likely at 100, 200 and 300 mM, since the seeds do not regain their maximum germination capacity. The body of data shows that salt inhibits the germination of *L. maritima* seeds mainly by an osmotic effect.

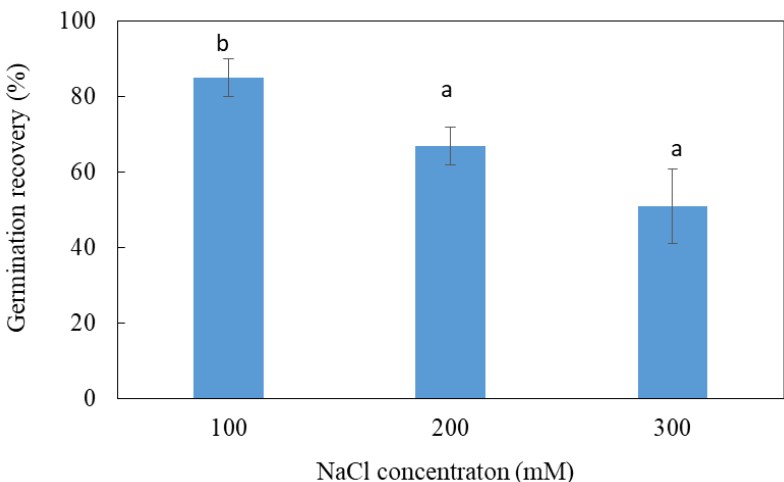

**Figure 2.** Germination recovery of ungerminated salt-treated seeds of *L. maritima*. Values at each level of salinity with the same letter are not significantly different ($p < 0.05$); Tukey test. Data are presented as means and standard errors of three repetitions.

### 3.3. Effects of Chemical Priming on Germination under NaCl Salinity

In the absence of salt, nitrate treatment accelerates germination (Figure 3A). Thus, after 20 days, the germination rate reaches 60% in the presence of nitrate against only 30% in its absence (Figure 4). Nitrogen treatment also increases the final germination: this parameter increases from 70% on control medium to 82% when nitrate is provided in the imbibition solution. The beneficial effect of nitrate on both components of germination (rate and final germination capacity) also appears at 50 mM NaCl, but this effect is much less pronounced than before. The beneficial effect of nitrate appears more significantly at 100 and 200 mM NaCl. This effect is exerted in a more pronounced way on the percentage than on the rate of germination. Thus, at these doses of salt, the first germinations are observed between 20 and 25 days in the presence of $NO_3^-$ against 30 days in the absence of $NO_3^-$. The final percentage of germination reaches 78% and 38%, respectively, at 100 and 200 mM for the first treatment. The beneficial action of $NO_3^-$ also appears at 300 mM NaCl since 10% of the seeds germinate against a zero germination rate in the absence of this nitrogenous source. In conclusion, $NO_3^-$ improves the germination capacity of *L. maritima* seeds, especially in the presence of salt.

Monitoring of germination kinetics in the presence of thiourea shows that the effect of this substance is dependent on the salinity of the medium (Figure 3B). The application of thiourea in the absence of salt does not significantly change the kinetics of germination. At 50 mM NaCl, the intake of thiourea slows down germination: the maximum germination rate is observed on the 35th day in the absence of this substance and on the 45th day in its presence. The final percentage of germination under salinity is also affected (Figure 4). At 100, 200 and 300 mM, thiourea improves germination capacity, which is manifested mainly by a shortening of the latency phase and particularly by an increase in germination capacity. This effect appears particularly at 200 mM NaCl, the final germination increases from about 5% in the absence of thiourea to 17% in its presence. There is also a partial lifting of dormancy induced by 300 mM NaCl. Overall, the effect of thiourea is reminiscent of that of $NO_3^-$ with an acceleration of germination and an increase in the percentage of sprouted seeds, particularly at 100 and to a lesser degree at 200 mM. Nevertheless, this beneficial action remains less pronounced than that of $KNO_3$.

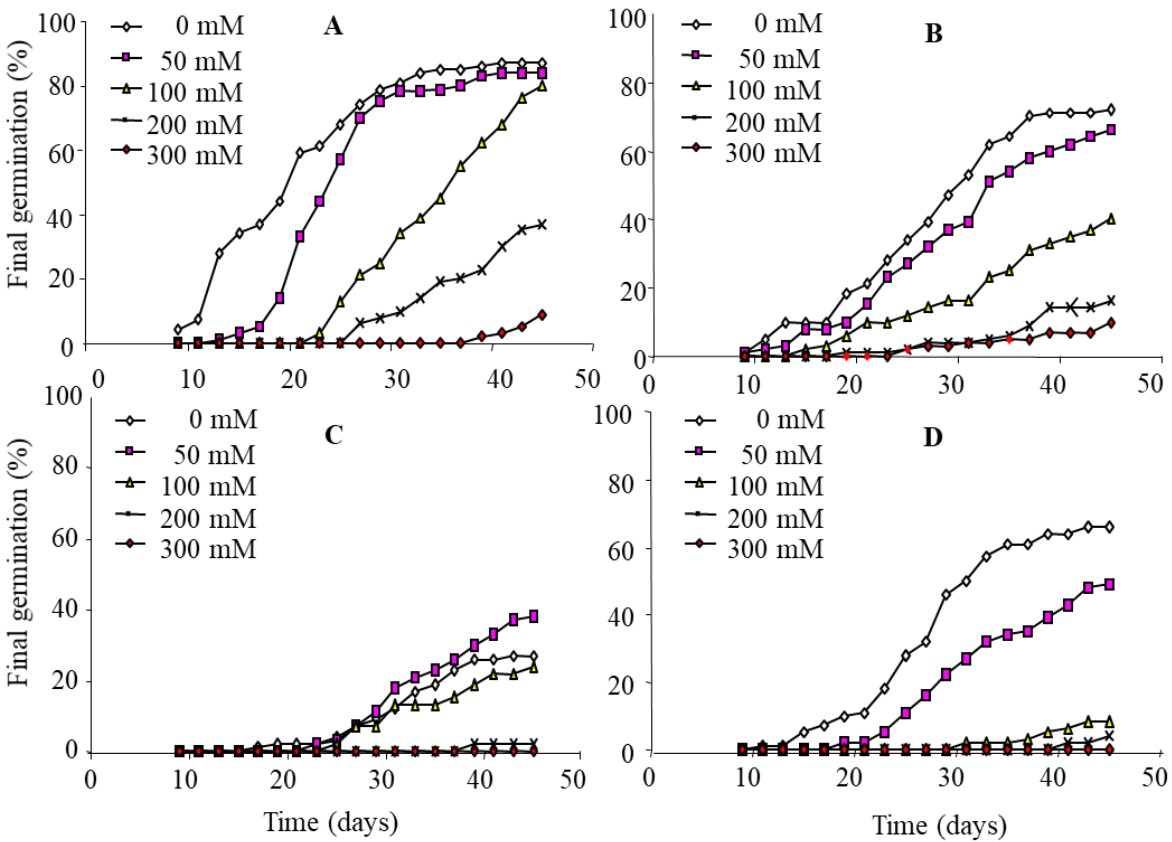

**Figure 3.** Effect of KNO₃ (**A**), thiourea (**B**), proline (**C**) and salicylic acid (**D**) pretreatment on the germination of *L. maritima*.

The application of proline at a dose of 1 mM lowers the germination rate and dramatically delays the germination process (Figure 3C). This action is noted for all doses of salt applied. Thus, after 45 days, the germination capacity reaches 27% in the absence of salt. Seeds subjected to 50 and 100 mM NaCl express a behavior close to that observed in the absence of salt. Since the germination process is still underway even after 45 days of germination, it is difficult to diagnose the nature of the depressive action of proline: is it a toxic or osmotic effect? The second type of effect would be unlikely since proline is given at a dose of 1 mM which will not significantly change the concentration of the imbibition solution which contains up to 200 mM NaCl. Moreover, we did not record an accentuation of the osmotic effect of salt by the intake of 20 mM of KNO₃ or 10 mM of thiourea. Given the unexpected effect of proline, several trials have been conducted and have confirmed the depressive action of this amino acid on germination.

In the absence of salt, salicylic acid does not appear to alter the kinetics of germination (Figure 3D). However, it accentuates the depressive action of salt. Indeed, we note a cancellation of the stimulatory effect of 50 mM NaCl with an increase in the duration of the latency phase and a decrease in the final germination from 77% in the absence of salicylic acid to 42% in its presence. At 100 and 200 mM NaCl, germination rates are also lower in the presence than in the absence of salicylic acid (Figure 4).

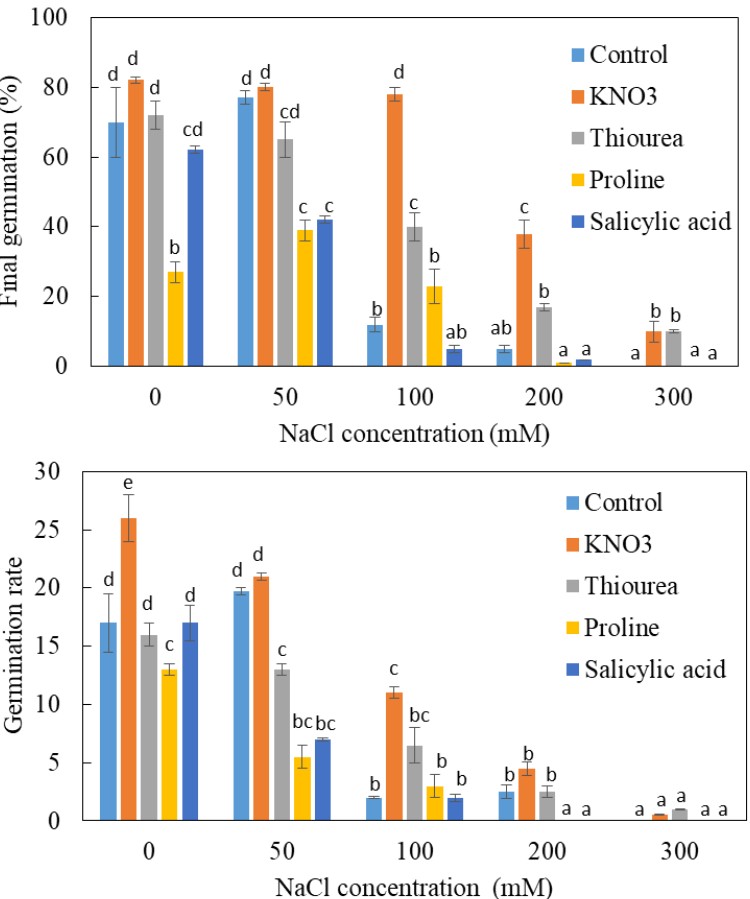

**Figure 4.** Comparative analysis of the final germination and germination rate of the unprimed (control) and primed seeds of *L. maritima*. Values at each level of salinity with the same letter are not significantly different ($p < 0.05$); Tukey test. Data are presented as means and standard errors of three repetitions.

## 4. Discussion

Salt (NaCl) does not affect the germination of *L. maritima* if applied at a moderate dose of 50 mM. Nevertheless, for higher concentrations of NaCl, there is a decrease in the germination rate or even a total inhibition of germination (200 and 300 mM). The depressive effect of NaCl on final germination and germination rate has been demonstrated in several halophytes. However, these plants maintain an ability to germinate (even at reduced rates) in a wide range of saline concentrations. Thus, the tolerance limits for salinity at the germination stage vary from one species to another. For example, the germination of *Urochondra setulosa* shows a considerable decrease of 80% at 300 mM NaCl [17]. The seeds of *Suaeda moquinii*, *Salicornia rubra*, *Halogeton glomeratus* and *Kochia scoparia* germinate weakly in the presence of 1000 mM [7]. Despite interspecific differences in germination salt sensitivity, it appears that most halophytes retain seed viability despite the high salt concentrations to which they are subjected during dormant periods. We noted that *L. maritima* shows rapid recovery after the transfer of ungerminated seeds to distilled water. The germination rate reaches 60% in seeds previously subjected to 300 mM. Similar behaviors have been described in *Halocnumum strobilacaeum*, *Salicornia ramosissima*, *Arthrocneuum macrostachyum*, *Sarcocornia fruticosa* [8], *Allenrolfea occidentalis* [7] and *Cakile maritima* [18]. Keiffer and Ungar [19] showed that the seeds of *Salicornia europaea* and *Suaeda calceoformis* regain a recovery rate that exceeds the rate observed with the control after a two-year saline pre-treatment. All these data confirm the osmotic nature of the effects of salt on halophyte germination. The seeds of these plants would control the accumulation of $Na^+$ and $Cl^-$ ions despite their abundance in the medium. This property, which ensures good protection of

seeds against the deleterious effects of salt, gives halophytes an ability to grow in biotopes characterized by fluctuations in soil salinity. Indeed, the absence of germination during the period of the year characterized by high salinity without affecting the viability of the seeds is a means of preserving the seeds until the conditions become favorable at ground level, during its desalination by rainfall. In order to improve the germination of *L. maritima* seeds at 100–300 mM NaCl, we studied the effect of priming the seeds with already known germination promoting substances such as asnitrate, thiourea, proline and salicylic acid on germination in the absence or presence of a range of concentrations of NaCl—50, 100, 200 and 300 mM.

Our results showed that nitrate improves the germination of *L. maritima* seeds, especially in the presence of salt. Stimulation of germination by $NO_3^-$ has been observed in many halophytes such as *Shoenia filifolia* [20], *Aconitum heterophyllum* [21], *Atriplex occidentalis* [22], *A. griffitii* and *Sporobolus arabicus* [23], *Paspalum vaginatum* [24] and *Crithmum maritimum* [12]. Other studies reported no effect of $KNO_3$ on *Crithumum maritmim*, French population [11], and *Silene mollissima* [25]. Despite the multitude of studies using $KNO_3$ to improve germination, the mode of action of $NO_3^-$ on germination is poorly understood. Several hypotheses have been proposed to explain this mechanism. The first experiments by Robert and Smith [26] hypothesized that nitrate activates the phosphate pentoses pathway, resulting in stimulation of germination. Subsequent studies have shown that this compound stimulates the absorption of oxygen necessary for seed dormancy [27]. A third hypothesis postulates that $NO_3^-$ acts as a cofactor of cytochromes [28]. Later, Hilhorst and Karssen [29] proposed a model that integrates the effect of temperature, nitrate and light on germination and dormancy. In this model, $NO_3^-$ binds to the plasma membrane and increases the affinity of the protein (phytochrome receptor) to the active form of phytochrome (Pfr). The formation of this complex (active receptor-form of phytochrome) induces transduction signals leading to the biosynthesis of gibberellic acid strongly involved in the mobilization of reserves, on which the germination process strongly depends. These data explain the stimulation of germination that we observed in control seeds soaked in $KNO_3^-$-enriched distilled water. Nevertheless, our results showed the beneficial effect of $NO_3^-$ especially in the presence of salt. In addition, the results of the first experiments showed that salt subjects seeds particularly to osmotic effects. Under these conditions, we believe that the addition of $NO_3^-$ in the imbibition solution would lead to a strong accumulation of this cation inside the cells to ensure an osmotic balance with the external environment, which is a favorable condition for imbibition and germination of seeds. Moreover, a strong positive relationship between the $NO_3^-$ content of seeds and their germination capacity has been observed in *L. maritima* (unpublished data).

In the present study, we noticed that thiourea increases the final germination of *L. maritima* seeds if applied at salt doses of 100, 200 or 300 mM but it does not increase the rate of germination. At this level, it appears less effective than potassium nitrate. Our results are thus in agreement with those of Khan and Ungar [23] obtained on the seeds of *Sporobolus arabicus* but are in contradiction with the data acquired by the same authors on the seeds of *Atriplex griffitii* or *Aconitum heterophyllum* [21]. The stimulation of germination induced by thiourea, especially in the presence of salt, reflects an ability of this nitrogen compound to alleviate the dormancy induced by the presence of NaCl in the imbibition solution. Indeed, for a long time, the effectiveness of nitrogen compounds in alleviating primary or secondary dormancy has been demonstrated in several plant species [30] by inducing gibberellin biosynthesis pathways. According to the same authors, thiourea is involved in the alleviation of dormancy in *Avena fatua* lines, however, it remains significantly less effective than $NO_3^-$ and its effect is strictly limited to the least dormant lines. The promoting action of thiourea on germination is related to its inhibitory effect on phenol oxidase [31], resulting in an increase in the availability of oxygen for enzymes expressing low affinity to this gas, such as the terminal oxidase of alternative respiration.

We have seen that proline, provided in the imbibition solution at the rate of 1 mM, alters the kinetics of germination and delays the germination process at all doses of the

salts used. On first examination, this result is unexpected given the role of proline in osmoregulation and its involvement in the conservation of cellular structures and their protection against reactive oxygen species [32]. In this context, several authors report an improvement in the germination of halophytes such as *Zygphyllum simplex*, *Allenophlora occidentallis* and *Atriplex griffithii* [7]. Other studies report a positive relationship between the ability of seeds to germinate and their ability to accumulate proline. For example, exposure of *Kosteletzkya virginika* seeds to salt during germination is accompanied by a marked increase in proline in sprouted seeds compared to ungerminated seeds [33]. However, this proline germination promoting action has not been observed in other plant species and under various experimental conditions. Thus, Khan and Ungar [9] noted that both proline and glycine-betaine partially alleviate the primary dormancy of *Zygophyllum simplex* seeds only when germination is conducted on distilled water or at very low salinities, not exceeding 25 mM NaCl. Both nitrogen compounds do not lead to any stimulation of germination under conditions of high salinity. Similarly, Khan and Ungar [9] noted that proline and betaine do not alleviate salt-induced dormancy in both types of seeds produced by *Arthrocnemum indicum*.

The inhibition of the germination of *L. maritima* seeds by proline, regardless of the salinity of the medium, would not be due to an osmotic effect. The intake of 1 mM of proline will not significantly alter the osmotic balance between the germinating seeds and their environment. One could rather think of a toxic effect of proline. The low hydration of the seeds that would be associated with a strong endocellular accumulation of proline would bring the concentration of this compound back to toxic levels. According to some studies, the accumulation of proline is not always a trait of tolerance. For example, in rice, the accumulation of proline is associated with symptoms of toxicity and is therefore a marker of sensitivity and not tolerance [34]. In the same context, Garcia et al. [35] noted that exogenous proline intake further develops these symptoms of toxicity in rice. Proline toxicity was also observed in transgenic plants of *Arabidopsis thaliana* [32]. In order to diagnose the effects associated with a high accumulation of proline, these authors produced transgenic plants deficient in proline dehydrogenase (PDH), an enzyme involved in the degradation of proline, by means of antisense constructions. The seeds from these transgenic plants are sown in a medium with the addition of increasing doses of proline. The authors noted an increased sensitivity of transgenic plants at the germination stage and post-germination growth. The authors have highlighted in particular a halt in the development of seedlings associated with cell mortality. They thus concluded that inhibition of the catabolic pathways of proline leads to toxic proline contents, up to 1068 µg of proline per g of fresh matter. In light of these data, we believe that a strong accumulation of proline in the germinating seeds of *L. maritima*, probably associated with limited activity of the amino acid's breakdown enzymes, would subject the seeds to toxic effects.

Salicylic acid was found to have no significant effect on the germination of *L. maritima* seeds at 0 mM NaCl. However, it seems to amplify the depressive effect of salt at higher salt concentrations. Our results are consistent with those obtained by Borsani et al. [36] on transgenic plants of *Arabidopsis thaliana*. By comparing the response of a wild lineage of *A. thaliana* to that of a transgenic line deficient in salicylic acid (integrating a trangen encoding salicylate hydrolase), these authors observed a particular sensitivity of the wild line subjected during germination or post-germination growth to 100 mM NaCl or 270 mM of mannitol. They conclude that salicylic acid enhances the effects of salinity and osmotic stress by increasing the genesis of reactive oxygen species during photosynthesis and germination in *A. thaliana*. However, the effect of salicylic acid seems to depend on its content in the tissues. Thus, several studies report a beneficial effect of salicylic acid on the response of plants to abiotic or biotic constraints. For example, incubation of *Raphanus sativus* seeds in a salicylic acid solution increases the optimal temperature range for germination [37]. The pre-treatment of maize seeds with salicylic acid induces antioxidant enzymes, resulting in an increase in its resistance to cold [38]. Salicylic acid improved the germination of *Leynus chinensis* under salt-alkali stress through an osmotic priming role [38].

## 5. Conclusions

Overall, it can be concluded that the germination of *L. maritima* seeds is improved by potassium nitrate in the presence or absence of salt, while thiourea increases the germination rate without affecting the germination rate. Salicylic acid amplifies the effect of salt, this effect is all the more important as the salinity of the imbibition solution increases. Whereas, proline delays germination without stopping it completely even after 46 days of treatment. These findings indicate that the application of KNO$_3$ and thiourea may be used to improve seed germination of *L. maritima*, which is of great interest for cultivating this plant for landscaping purposes of saline soils.

**Author Contributions:** K.B.H.: Conceptualization, data curation, writing—original draft preparation, writing—review and editing, figures, supervision, project coordination. I.Z., A.D. and S.Y.: Data curation, writing—original draft preparation, figures. All authors have read and agreed to the published version of the manuscript.

**Funding:** Financial support of the project "HaloFarMs" by PRIMA, a program supported by the European Union, and by the Ministry of Higher Education and Scientific Research (MHES), Tunisia.

**Institutional Review Board Statement:** Not applicable.

**Informed Consent Statement:** Not applicable.

**Data Availability Statement:** Not applicable.

**Conflicts of Interest:** The authors declare no conflict of interest.

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
