# Peer review of "Effects of Chemical Priming on the Germination of the Ornamental Halophyte Lobularia maritima under NaCl Salinity"

_2674-1024, doi:10.3390/seeds1020009_

Round 1
Reviewer 1 Report
Dear Authors,
The manuscript provides new information about the effects of chemical priming on germination of the ornamental halophyte Lobularia maritima under NaCl salinity. The topic of manuscript is current, significant and interdisciplinary. The format of manuscript is adequate and usually follows the journal style including the text, tables, figures and references. I recommend this article for publication with minor revision.
Abstract
The length of abstract is appropriate. The abstract is compact, establishes the topic and summarizes the most important results of research.
Keywords: Please, avoid in keywords same words as in title
- Materials and methods
Line 89: These images are unnecessary. I suggest deleting them.
Line 90-109: Points from 2.2 to 2.3.2 should be reorganized. This point should have one subheading "Seed germination test (bioassay)". First, you need to write the source where the seed material was collected; Which treatments were applied and in which conditions; Which parameters were measured;
Line 110-124: Formulas should be added in a single paragraph, to be clearer.
Results contain valuable new information and discuss the author’s results with the adequate, up-to-date literature properly. Authors evaluated their results critically from many respects.
Discussion
Line 200-371: This section is too wordy. I suggest that the discussion part should be in one paragraph, without subheadings. Also, when you shorten this section, the number of references will be reduced.
The language and style of manuscript adequate, scientific at the same time easily comprehensible by a non-native reader as well.
Please, see comments in the pdf document.

Author Response
The manuscript provides new information about the effects of chemical priming on germination of the ornamental halophyte Lobularia maritima under NaCl salinity. The topic of manuscript is current, significant and interdisciplinary. The format of manuscript is adequate and usually follows the journal style including the text, tables, figures and references. I recommend this article for publication with minor revision.
Abstract
The length of abstract is appropriate. The abstract is compact, establishes the topic and summarizes the most important results of research.
Keywords: Please, avoid in keywords same words as in title
Response: Done
Materials and methods
Line 89: These images are unnecessary. I suggest deleting them.
Response : Done
Line 90-109: Points from 2.2 to 2.3.2 should be reorganized. This point should have one subheading "Seed germination test (bioassay)". First, you need to write the source where the seed material was collected; Which treatments were applied and in which conditions; Which parameters were measured;
Response : The M&M section was revised accordingly
Line 110-124: Formulas should be added in a single paragraph, to be clearer.
Response : Done
Results contain valuable new information and discuss the author’s results with the adequate, up-to-date literature properly. Authors evaluated their results critically from many respects.
Response : None
Discussion
Line 200-371: This section is too wordy. I suggest that the discussion part should be in one paragraph, without subheadings. Also, when you shorten this section, the number of references will be reduced.
Response : Done accordingly
The language and style of manuscript adequate, scientific at the same time easily comprehensible by a non-native reader as well.
Response : None
Please, see comments in the pdf document.
Response : Thank you
Reviewer 2 Report
The laboratory experiments were carried out in 2019 to assess the effects of salinity on Lobularia maritima seed germination and on germination recovery from the effects of saline conditions after transfer to distilled water. The manuscript is clearly written and has the correct structure. The topic matches the journal’s scope.
The introduction provides a good, generalized background. Establishes the originality of the research aims by demonstrating the need for investigations in the topic area. The objective is clearly defined. This is followed by a detailed description of the materials and methods.
The presented data is free from obvious errors. The results are concise and are separated from the discussion section. Results are well referenced with experimental data summarized in 3 figures. Statistical methods have been described with sufficient detail and were correctly applied. The discussion explores the significance of the results in the context of relevant published papers. The literature cited in the work is relevant to the study. The paper refers to 61 prior art publications, all of which are reasonably well referenced in the text. The conclusions of the study are presented in a short section highlighting the key results and their significance.
Author Response
The laboratory experiments were carried out in 2019 to assess the effects of salinity on Lobularia maritima seed germination and on germination recovery from the effects of saline conditions after transfer to distilled water. The manuscript is clearly written and has the correct structure. The topic matches the journal’s scope.
The introduction provides a good, generalized background. Establishes the originality of the research aims by demonstrating the need for investigations in the topic area. The objective is clearly defined. This is followed by a detailed description of the materials and methods.
The presented data is free from obvious errors. The results are concise and are separated from the discussion section. Results are well referenced with experimental data summarized in 3 figures. Statistical methods have been described with sufficient detail and were correctly applied. The discussion explores the significance of the results in the context of relevant published papers. The literature cited in the work is relevant to the study. The paper refers to 61 prior art publications, all of which are reasonably well referenced in the text. The conclusions of the study are presented in a short section highlighting the key results and their significance.
Response : We appreaciated your positive comments.
Reviewer 3 Report
The study deals with the preparation of Lobularia martina seed material, which in natural conditions inhabits areas with increased soil salinity. Getting to know the specificity of the germination of the analyzed seeds is important primarily in terms of shaping green areas in places where there is access, for example, to sea water, in the context of saving drinking water resources. The work is valuable, and the Discussion section tries to describe the reported relationships in detail, although in my opinion too much information was given about the nature of processes that were not directly investigated in this paper. The main reservations relate to the imprecise description of the research methods used, which would make it difficult to repeat the experiment if necessary.
In the Keywords section, as already mentioned, you should use synonyms for the forms used in the title and not repeat them. It is about a greater possibility of finding this article in various databases when looking for information on the subject matter covered in this paper.
In accordance with the journal's requirements contained in the Microsoft Word template file, a numerical citation system should be used. Thanks to this, the authors could discover that 2 items cited in the text were not included in the References section, and as many as 15 items from the References section are not referenced in the text of the manuscript. In addition, many sources were published in the previous century and I have the impression that at least some of them could be replaced with more recent literature.
In the Plant materials subsection, I propose to refer to a literature source when describing post-harvest treatment of seeds. It is not known what the parameters of this procedure were based on. The details of obtaining seeds from plants were also not provided. Figure 1 shows a view of the plants of the species under study, which bloomed in May, so probably no seeds were collected from them at that time.
The authors report that the germination effects were assessed for 7 consecutive days, and the results only present data after this period. Even considering that only non-germinating seeds were taken into account for further analysis, the results of this first stage should be presented. In my opinion, in the light of the results obtained, the description of the handling of seeds is imprecise and it is not entirely clear how the batches were handled. Figure 2 shows that the germination assessment was carried out for 45 days. Was it only after this period that the seeds were considered to be ungerminated? What about germination of seeds treated with distilled water for 7 days (lines 100-102)?
On what basis were the concentrations of chemical solutions given in the study assumed? Does this result from the already adopted methodology for conducting such treatments, or did the authors choose these parameters on their own? Perhaps the use of other concentrations of the analyzed preparations would be more beneficial for the germination process of seeds of this species. Why in this case the temperature of the germination process was set at a higher level, 25°C? If, as described here, only seeds kept at 0, 200 and 300 mM NaCl concentrations were considered, why is Figure 4 also shown 50 and 100 mM, ignoring 150 mM.
I propose to save the mathematical dependencies given in the work using the equation editor.
The manufacturer of the statistical software used must also be reported.
Figure 2 does not present the germination curve for the concentration of 150 mM NaCl, but its description is included in the text (similar to Figure 4). No mention is made of the lack of seed germination at a concentration of 300 mM NaCl (this variant is also omitted in further descriptions). Why do the values of the germination rates given in the text do not correspond to those read in the figure?
Figure 3 does not include the standard errors value. It can be stated, as before, that the values given in the text differ slightly from those presented in the diagram.
From the last paragraph of the Discussion section, I propose to separate the Conclusions section.
There are no entries in the following subsections, eg Author Contributions.
The bibliographic data entries in the References section were not prepared in accordance with the journal guidelines.
In addition, the authors should note the following minor shortcomings:
- lines 3, 8, 23, 280-1, References section – Latin names of the mentioned species should be written in italics,
- when citing publications of several authors, the abbreviation "et al." Is used - there is a large inconsistency in writing this form (with or without a dot, sometimes with the use of a comma),
- lines 57 and 60 – remove unnecessary brackets,
- line 63 – the correct spelling of the name is: Magné,
- line 70 – the correct spelling of the name is: Picó,
- the correct record of a chemical compound is KNO3; also remember to change the form of recording the ions of this compound,
- lines 209, 213 and 241 – should be: Gulzar and Khan 2001, unless it is a reference not listed in the References section,
- lines 218 and 256 – no spaces,
- line 228 – check the year of publication (different in the References section),
- lines 223 and 314 – remove unnecessary letter indices for the year of source publication,
- lines 258 and 357 – error in the name of the author of the publication,
- line 278 – no dot,
- lines 317 – should be: Gulzar and Khan (1998), unless it is a reference not listed in the References section.
Author Response
The study deals with the preparation of Lobularia martina seed material, which in natural conditions inhabits areas with increased soil salinity. Getting to know the specificity of the germination of the analyzed seeds is important primarily in terms of shaping green areas in places where there is access, for example, to sea water, in the context of saving drinking water resources. The work is valuable, and the Discussion section tries to describe the reported relationships in detail, although in my opinion too much information was given about the nature of processes that were not directly investigated in this paper. The main reservations relate to the imprecise description of the research methods used, which would make it difficult to repeat the experiment if necessary.
Response : The Mn was revised according to your comments
In the Keywords section, as already mentioned, you should use synonyms for the forms used in the title and not repeat them. It is about a greater possibility of finding this article in various databases when looking for information on the subject matter covered in this paper.
Response : Done
In accordance with the journal's requirements contained in the Microsoft Word template file, a numerical citation system should be used. Thanks to this, the authors could discover that 2 items cited in the text were not included in the References section, and as many as 15 items from the References section are not referenced in the text of the manuscript. In addition, many sources were published in the previous century and I have the impression that at least some of them could be replaced with more recent literature.
Response : Revised accordingly
In the Plant materials subsection, I propose to refer to a literature source when describing post-harvest treatment of seeds. It is not known what the parameters of this procedure were based on. The details of obtaining seeds from plants were also not provided. Figure 1 shows a view of the plants of the species under study, which bloomed in May, so probably no seeds were collected from them at that time.
Response : Revsied accordingly
The authors report that the germination effects were assessed for 7 consecutive days, and the results only present data after this period. Even considering that only non-germinating seeds were taken into account for further analysis, the results of this first stage should be presented. In my opinion, in the light of the results obtained, the description of the handling of seeds is imprecise and it is not entirely clear how the batches were handled. Figure 2 shows that the germination assessment was carried out for 45 days. Was it only after this period that the seeds were considered to be ungerminated? What about germination of seeds treated with distilled water for 7 days (lines 100-102)?
Response : The description of the M&M was corrected according to your comments.
On what basis were the concentrations of chemical solutions given in the study assumed? Does this result from the already adopted methodology for conducting such treatments, or did the authors choose these parameters on their own? Perhaps the use of other concentrations of the analyzed preparations would be more beneficial for the germination process of seeds of this species. Why in this case the temperature of the germination process was set at a higher level, 25°C? If, as described here, only seeds kept at 0, 200 and 300 mM NaCl concentrations were considered, why is Figure 4 also shown 50 and 100 mM, ignoring 150 mM.
Response : See revised M&M section
I propose to save the mathematical dependencies given in the work using the equation editor.
Response : Done
The manufacturer of the statistical software used must also be reported.
Response : Done
Figure 2 does not present the germination curve for the concentration of 150 mM NaCl, but its description is included in the text (similar to Figure 4). No mention is made of the lack of seed germination at a concentration of 300 mM NaCl (this variant is also omitted in further descriptions). Why do the values of the germination rates given in the text do not correspond to those read in the figure?
Response : You are right. We corrected te values and revised these results accordingly
Figure 3 does not include the standard errors value. It can be stated, as before, that the values given in the text differ slightly from those presented in the diagram.
Response : Done
From the last paragraph of the Discussion section, I propose to separate the Conclusions section.
Response : Done
There are no entries in the following subsections, eg Author Contributions.
Response : Entries added
The bibliographic data entries in the References section were not prepared in accordance with the journal guidelines.
Response : Revised accordingly
In addition, the authors should note the following minor shortcomings:
lines 3, 8, 23, 280-1, References section – Latin names of the mentioned species should be written in italics,
when citing publications of several authors, the abbreviation "et al." Is used - there is a large inconsistency in writing this form (with or without a dot, sometimes with the use of a comma),
lines 57 and 60 – remove unnecessary brackets,
line 63 – the correct spelling of the name is: Magné,
line 70 – the correct spelling of the name is: Picó,
the correct record of a chemical compound is KNO3; also remember to change the form of recording the ions of this compound,
lines 209, 213 and 241 – should be: Gulzar and Khan 2001, unless it is a reference not listed in the References section,
lines 218 and 256 – no spaces,
line 228 – check the year of publication (different in the References section),
lines 223 and 314 – remove unnecessary letter indices for the year of source publication,
lines 258 and 357 – error in the name of the author of the publication,
line 278 – no dot,
lines 317 – should be: Gulzar and Khan (1998), unless it is a reference not listed in the References section.
Response: Thank you. All these shortcomings with other are considered, checked and corrected in the revised Mn.